# PEDF Deletion Induces Senescence and Defects in Phagocytosis in the RPE

**DOI:** 10.3390/ijms23147745

**Published:** 2022-07-13

**Authors:** Ivan T. Rebustini, Susan E. Crawford, S. Patricia Becerra

**Affiliations:** 1Section of Protein Structure and Function, Laboratory of Retinal Cell and Molecular Biology, National Eye Institute, National Institutes of Health, Bethesda, MD 20892, USA; ivan.rebustini@nih.gov; 2Department of Surgery, North Shore University Research Institute, University of Chicago Pritzker School of Medicine, Chicago, IL 60201, USA; crawford1@uchicago.edu

**Keywords:** PEDF, phagocytosis, RPE, *Serpinf1*, senescence

## Abstract

The retinal pigment epithelium (RPE) expresses the *Serpinf1* gene to produce pigment epithelium-derived factor (PEDF), a retinoprotective protein that is downregulated with cell senescence, aging and retinal degenerations. We determined the expression of senescence-associated genes in the RPE of 3-month-old mice that lack the *Serpinf1* gene and found that *Serpinf1* deletion induced *H2ax* for histone H2AX protein, *Cdkn1a* for p21 protein, and *Glb1* gene for β-galactosidase. Senescence-associated β-galactosidase activity increased in the *Serpinf1* null RPE when compared with wild-type RPE. We evaluated the subcellular morphology of the RPE and found that ablation of *Serpinf1* increased the volume of the nuclei and the nucleoli number of RPE cells, implying chromatin reorganization. Given that the RPE phagocytic function declines with aging, we assessed the expression of the *Pnpla2* gene, which is required for the degradation of photoreceptor outer segments by the RPE. We found that both the *Pnpla2* gene and its protein PEDF-R declined with the *Serpinf1* gene ablation. Moreover, we determined the levels of phagocytosed rhodopsin and lipids in the RPE of the *Serpinf1* null mice. The RPE of the *Serpinf1* null mice accumulated rhodopsin and lipids compared to littermate controls, implying an association of PEDF deficiency with RPE phagocytosis dysfunction. Our findings establish PEDF loss as a cause of senescence-like changes in the RPE, highlighting PEDF as both a retinoprotective and a regulatory protein of aging-like changes associated with defective degradation of the photoreceptor outer segment in the RPE.

## 1. Introduction

The retinal pigment epithelium (RPE) is the main source of pigment epithelium-derived factor (PEDF) for the retina [1]. PEDF is an important protein that contributes to the homeostasis of the retina and RPE. Monolayers of polarized RPE cells express the *SERPINF1* gene for PEDF and release the glycoprotein in an apicolateral fashion into the interphotoreceptor matrix, where it acts on cell survival and avascularization [2]. However, RPE production and secretion of PEDF decline with aging [3,4,5,6], as well as with progression of retinal degenerations and RPE damage [7]. Similarly, its expression declines in other tissues during aging in vivo, such as in skin [8], and during senescence of cells in vitro, such as in WI-38 lung fibroblast in which the levels of transcripts and secreted PEDF protein are >100-fold lower than in young cells [9]. These observations suggest that depletion of PEDF associates with senescence, aging and its consequences in age-related diseases.

PEDF is a member of the serine protease inhibitor (serpin) superfamily [10] that protects retinal neurons, photoreceptors and RPE from pathological damage [11], as well as preventing retinal and choroidal neovascularization [12,13], and therefore has been likened to an ‘ocular guardian’ [14,15]. This serpin mediates its activities by interactions with binding partners rather than by its potential inhibition of serine proteases. One of the binding partners is the PEDF receptor PEDF-R. PEDF-R is a product of the *PNPLA2* (patatin-like phospholipase A2) gene that is expressed in the retina and RPE [16] and is a key gene for lipid metabolism. PEDF-R mediates the survival and neurotrophic activities of PEDF in retinal cells. The fact that PEDF binds and stimulates the PEDF-R lipase activities [17] has placed PEDF as a regulator of lipid metabolism.

Increasing lines of evidence suggest that lipid metabolism and lipid reprogramming play an important role in aging, longevity, and age-related diseases [18,19]. In this regard, lipid-related gene variants associated with human aging, and lipid biomarkers of aging in humans have been identified [20]. There is also precedent that targeting genes involved in lipid hemostasis affects aging in animal models [19,20,21]. Plasma lipidomic profiles of 11 different mammalian species with longevities varying from 3.5 to 120 years can accurately predict the lifespan of animals and, in particular, plasma long-chain free fatty acids and lipid peroxidation-derived products are inversely correlated with longevity [22]. Similarly, the genomes of 25 different species reveal that genes involved in lipid composition undergo increased selective pressure in longer-lived animals [23]. Evidence from animals with extreme longevity also links lipid metabolism to aging. The ocean quahog clam *Arctica islandica*, an exceptionally long-lived animal that can survive for more than 500 years, exhibits a unique resistance to lipid peroxidation in mitochondrial membranes [24]. The bowhead whale, another complex animal with extreme longevity, which can live longer than 200 years, has lens membranes that are especially enriched with phospholipids. This unique enrichment is thought to partially underlie its uncanny resistance to the age-related lens disease of cataracts [25]. Naked mole rats, which enjoy remarkably long lifespans for rodents, have a unique membrane phospholipid composition that has been theorized to contribute to their exceptional longevity [26]. The importance of lipids in lifespan is further confirmed by the ability of lipid-related interventions to enhance longevity in model organisms [27]. It has been reported that senescent cells accumulate lipid droplets and import lipid tracers and that treating proliferating cells with specific lipids induces senescence, placing deregulation of lipid metabolism as a factor to regulate cellular senescence [20].

There are also pathologies and conditions associated with altered nuclear shape in which lipid modification of lamin A causes premature aging. Normal aging is also associated with abnormal nuclear shape, and linked to lipid modifications of nuclear lamin, and progerin [28,29]. Interestingly, nuclear shape can be affected by lipid synthesis. For example, inactivation of lipid phosphatase (lipin) causes expansion and alteration of nucleus phospholipid synthesis and an expanding nucleus [28,29]. Moreover, aging and progression of age-related macular degeneration leads to inefficient RPE phagocytosis and eventually retinal degeneration [30,31]. Age-related changes have been described in the RPE of rats, such as lipid accumulation in the aging RPE, but are quite rare in the RPE of 4- and 11-month-old animals [32]. At the same time, phagocytosis of rod outer segment discs decreases from 4 to 32 months of age [32]. Our previous findings, that *Pnpla2* downregulation delays the rhodopsin and lipid digestion in the RPE undergoing phagocytosis, imply that the efficiency of phagocytosis depends on PEDF-R [33], potentially linking PEDF in the process. However, the participation of PEDF in phagocytosis and these PEDF-R mediated activities is unknown.

These results raise the interesting question of whether PEDF may contribute to aging prevention in the RPE. In this study, we used a *Serpinf1* null mouse to address these questions.

## 2. Results

### 2.1. Serpinf1 Deletion Increased Senescence-Associated Gene Expression

To explore the effects of PEDF-deficiency in the RPE, we selected the *Serpinf1^−/−^* mouse. Genotyping of the *Serpinf1^−/−^* confirmed the deletion of the *Serpinf1* gene in these mice (Figure 1A). The PCR products migrated in the agarose gels as expected for the *Serpinf1^+/+^* mice (DNA fragments of 490 base pair (bp) band), and for *Serpinf1^−/−^* null mice (a DNA fragment of 650 bp) (Figure 1A). Given our interest in the effects of PEDF ablation on the RPE, we determined the levels of *Serpinf1* transcript and PEDF protein in the RPE used in this study. Production of transcripts followed by qPCR further confirmed that these mice had RPE that was free of *Serpinf1* mRNA. RPE *Serpinf1* transcripts were undetectable in the *Serpinf1^−/−^* and decreased by 67% in the *Serpinf1^+/−^* relative to those from littermate controls (Figure 1B). We further confirmed the depletion of PEDF protein in the RPE of the *Serpinf1^−/−^* mice by Western blot (Figure 1C). A single PEDF-immunoreactive band migrating as a protein of expected size for PEDF (50-kDa) was detected only for *Serpinf1^+/+^* and *Serpinf1^+/−^* RPE in the gels. These observations indicated a successful deletion of the *Serpinf1* gene leading to no transcription in the RPE, and that PEDF protein was depleted in the RPE of the *Serpinf1^−/−^* mice.

Focusing on the effects of PEDF deficiency on the aging RPE, we determined the expression of selected senescent-associated genes in the RPE of the *Serpinf1^−/−^* mouse. We used cDNA prepared from RNA extracted from freshly dissected *Serpinf1^+/+^*, *Serpinf1^+/−^* and *Serpinf1^−/−^* RPE/choroid eyecups, to perform quantitative PCR (qPCR). The following genes have previously been associated with senescence and therefore were selected for this study: *H2ax* (histone 2ax) [34], *Trp53* (or P53) [35], *Cdkn1a* (or P21) [36] and *Glb1* (β-galactosidase) [37]. Figure 2 shows that transcription of the *H2ax*, *Cdkn1a* and *Glb1* genes increased in the *Serpinf1^−/−^* RPE, but no increase was observed for the *Trp53* gene. The *H2ax* levels of the *Serpinf1^+/−^* and *Serpinf1^−/−^* RPE were of 1.3-fold and 1.6-fold, respectively, of the *Serpinf1^+/+^* littermate control. The *Cdkn1a* levels of the *Serpinf1^+/−^* and *Serpinf1^−/−^* RPE were both 1.3-fold of the *Serpinf1^+/+^* littermate control. Finally, the *Glb1* levels of the *Serpinf1^+/−^* and *Serpinf1^−/−^* RPE were 1.1-fold and 1.6-fold, respectively, of the *Serpinf1^+/+^* littermate control.

### 2.2. Serpinf1 Deletion Increased Β-Galactosidase Activity in the RPE

The increase in *Glb1* led us to investigate the senescence-associated β galactosidase (SA-β-gal) activity because the enzyme catalyzes the hydrolysis of β-galactosides into monosaccharides predominantly in lysosomes of senescent cells and is regarded a biomarker of cellular senescence [38]. The in-situ reaction in the flat mounts showed that the RPE cells of *Serpinf1^−/−^* mice (Figure 3A) had accumulated more fluorescent 5-DAF (a product from the enzymatic activity of SA-β-gal on its substrate C12FDG) than had the RPE cells of the control animals (Figure 3B). To quantitatively evaluate the effect of PEDF removal on the SA-β-gal activity, the intensity of 5-DAF fluorescence per RPE cell was determined in both genotypes and plotted. The plot of the relative 5-DAF fluorescence per RPE cell showed that the *Serpinf1^−/−^* RPE exhibited SA-β-gal activity 2.2-fold than that of the wild-type RPE (Figure 3C).

Together, the above observations indicated that *Serpinf1* and PEDF deficiency induced senescence-like changes, namely of senescence-associated gene expression and SA-β-gal activity, in the RPE.

### 2.3. Serpinf1 Deletion Increased the Size of Nuclei of the RPE Cells

Given that senescent cells undergo nuclear size changes [39,40], we examined the cellular morphology of the *Serpinf1^−/−^* RPE by fluorescent confocal microscopy. RPE/choroid flat mounts were prepared and stained to detect nuclei and F-actin. We observed that the RPE cells of the *Serpinf1^−/−^* mice had a slightly more intense F-actin staining than in the *Serpinf1^+/+^* littermate controls. More interestingly, the nuclei of *Serpinf1^−/−^* RPE were larger than the ones of the control mice (Figure 4A,B, respectively). To quantitatively assess the effect of PEDF removal on RPE nuclear size, the volumes of the nuclei were determined from their diameter—assuming round shaped nuclei- in both genotypes and were plotted. The plot showed that the *Serpinf1^−/−^* mice possess larger nuclei, being 2.9-fold greater than ones of the *Serpinf1^+/+^* mice (Figure 4C). We also noticed that the number of nucleoli per nucleus in the *Serpinf1^−/−^* RPE cells (8.2 ± 1.2) was 2-fold that of *Serpinf1^+/+^* RPE cells (4.2 ± 1.4). Additionally, the *Serpinf1^+/+^* RPE revealed a highly organized F-actin distribution pattern along the cytoplasmic rim of the cell. It is characterized by two parallel lines and repetitive condensed bands of actin filaments with a circular-like configuration at the confluence point of multiple cells. In striking contrast, *Serpinf1^−/−^* RPE showed a disorganized distribution pattern of F-actin with most cells demonstrating only a single linear line and loss of condensed regions of actin. In some areas, there appears to be disassembly of F-actin with multiple foci of clumped positively-stained material along the perimeter of the cell and extending within the cytoplasm. Unlike WT controls, there is marked heterogeneity in the size and shape of the cells. These findings suggested that PEDF deletion affected not only the morphology of RPE cells, but their nuclei size and nucleoli number implying that they are undergoing chromatin remodeling.

### 2.4. Serpinf1 Ablation Downregulated Pnpla2 in the RPE

The above findings suggest that the senescence-like changes caused by *Serpinf1* deficiency may impair RPE functions, such as phagocytosis-related events. Recently we have shown that degradation of photoreceptor outer segments by the RPE requires PEDF-R, and that PEDF-R downregulation delays POS digestion during phagocytosis [33]. First, we determined the levels of *Pnpla2* expression. Figure 5A shows that the levels of RPE *Pnpla2* transcripts decreased by 48% in *Serpinf1^+/−^* and 56% in *Serpinf1^−/−^*, when compared with control *Serpinf1^+/+^* mice of the same age as *Serpinf1^−/−^*. Figure 5B shows that PEDF-R protein levels also decreased by 35.3% when compared to the *Serpinf1^+/+^* RPE. These observations indicated that *Serpinf1* gene deficiency associates with a concerted decline in gene expression of its binding partner PEDF-R in the RPE, and in turn it could negatively affect photoreceptor outer segments digestion.

### 2.5. Rhodopsin and Lipid Accumulated in the Serpinf1-Depleted RPE

Second, we investigated the rhodopsin turnover and lipid accumulation in the RPE, two post-phagocytic activities [41,42]. The RPE/choroid flat mounts were prepared from *Serpinf1^−/−^* and *Serpinf1^+/+^* mice at 5–7 h after light onset, and were stained to detect rhodopsin (RHO), F-actin, and nuclei. We observed again that the F-actin fluorescence in the RPE of *Serpinf1^−/−^* mice was more intense than in controls *Serpinf1^+/+^*. More interestingly, the RHO particles of the *Serpinf1^−/−^* mice (Figure 6A) were more abundant than in the control *Serpinf1^+/+^* mice (Figure 6B). To quantitatively assess the effect of PEDF deficiency on rhodopsin turnover, the number of particles per RPE cell was determined for each genotype and plotted. The plot of the number of rhodopsin-stained particles per cell showed that the *Serpinf1^−/−^* RPE accumulated RHO 3.5-fold than in the *Serpinf1^+/+^* RPE (Figure 6C).

Similarly, the RPE flat mounts were prepared from *Serpinf1^−/−^* and *Serpinf1^+/+^* mice at 5–7 h after light onset to detect lipid with BODIPY, F-actin and nuclei. We observed that the lipid deposits of the *Serpinf1^−/−^* mice (Figure 7A) were more abundant and larger than in the control mice (Figure 7B). To quantitatively assess the effect of PEDF deficiency on lipid accumulation in RPE, the intensity of BODIPY per region of interest (ROI) was determined for each genotype and plotted. The plot of BODIPY fluorescence showed that the *Serpinf1^−/−^* RPE accumulated lipid deposits 2-fold than in the *Serpinf1^−/−^* RPE (Figure 7C).

Altogether, these findings showed that absence of PEDF negatively affected the turnover of the phagocytosed rhodopsin and lipids by RPE cells, implying a defective digestion of photoreceptor outer segment tips, and that the efficiency of RPE phagocytosis also depends on PEDF.

## 3. Discussion

Here we show that the loss of PEDF causes senescence-like changes in the RPE. This conclusion is supported by the observed (1) induction of senescence-related genes *H2ax*, *Cdkn1a*, and *Glb1*, (2) increases of the senescence-associated β-galactosidase activity, (3) nuclear enlargement, and (4) disorganized distribution pattern of F-actin and other phenotypic changes in the RPE of *Serpinf1^−/−^* mice when compared with wild-type RPE. In addition, the fact that the *Serpinf1^−/−^* RPE downregulates the phagocytosis-related *Pnpla2* gene and accumulates rhodopsin and lipids points to RPE phagocytosis dysfunction caused by PEDF removal. Our findings agree with previous demonstrations of RPE phagocytic function decline with aging [30]. Together they provide evidence that PEDF contributes to both senescence prevention and phagocytosis, highlighting its role as double agent in supporting the RPE function and its consequences.

This is the first time that the *Serpinf1^−/−^* mouse has been used to study senescence and phagocytosis of the RPE. Previously we used this mouse to investigate the impact of PEDF deficiency on the neural retina and reported that PEDF deficiency increases the susceptibility of *rd10* mice to retinal degeneration [16]. In that study, while no remarkable differences were seen between the RPE of the *Serpinf1^−/−^* and *Serpinf1*^+/+^ mice except for regions of irregular basal infolds, we observed many degenerative vacuoles in some RPE cells by TEM. Interestingly, histology revealed swollen and less pigmented RPE cells in the *rd10*/*Serpinf1^−/−^* mice than in *rd10*, with disorganized ONL, suggesting a causal link with lack of PEDF. In the present study, the nuclear enlargement and lipid and rhodopsin particle accumulation in *Serpinf1^−/−^* RPE flat mounts concur with swollen RPE. In various diseases, as well as in aging, nuclear shape is altered [28]. For example, normal aging is associated with abnormal nuclear shape, and an aberrant form of the nuclear lamina component Lamin-A called Progerin, which is constitutively lipid-modified causes the premature aging syndrome Hutchison-Gilford progeria [43,44]. Moreover, there are lines of evidence that nuclear shape is affected by phospholipid metabolism, e.g., inactivation of lipid phosphatase (lipin) causes expansion and alteration of the nucleus [45]. It is not known whether PEDF/PEDF-R can directly or indirectly affect nuclear size by their involvement in lipid metabolism.

The absence of *Serpinf1* expression in the ONL of *Serpinf1*^+/+^ mice implies that the photoreceptors must rely on other cells, such as their neighboring RPE, for PEDF production and supply to act on survival. However, as PEDF declines with senescence or aging, the RPE becomes dysfunctional lowering its capacity to digest POS and the retina becomes permissive to degeneration. Hence, in this way PEDF is a key contributor to RPE homeostasis.

From this study, we postulate that PEDF is required to enhance the activity of PEDF-R in the RPE to promote digestion of the phagocytosed POS tips by a yet unknown molecular mechanism. A possible mechanism of action of PEDF signaling in the RPE consists of the involvement of PEDF-R lipase in digesting the phospholipids of the POS to expose rhodopsin to proteases upon PEDF binding. We have demonstrated that PEDF-R is located at the cell surface of RPE cells and that the extracellular PEDF binds to it [46]. PEDF-R has been detected surrounding lipid droplets in adipocytes [47] and hepatocytes [48]. Accordingly, PEDF may act on RPE via PEDF-R to promote phagocytosis-related events.

Senescent cells accumulate with age and contribute to the normal aging process as well as age-related macular degeneration [49]. Aging and age-related diseases in humans have been linked to shortening of the telomere, a unique DNA-protein complex which covers the ends of chromosomes to avoid end fusion and maintain their stability and integrity. Interestingly, knockdown of *PNPLA2* shortened the telomere length in human liver HepG2 cells, implying the association of lipid metabolism impairment with telomere shortening and aging [50] and suggesting a PEDF/PEDF-R link with preservation of telomere length and anti-aging.

In conclusion, our study provides evidence of the importance of PEDF signaling in age-related disease processes. Utilization of PEDF deficient mouse models can provide mechanistic insight in studies of senescence and aging and can help delineate new signaling pathways involving phagocytosis that impact lipid and visual pigment recycling in ocular pathologies.

## 4. Materials and Methods

### 4.1. Animals

Littermates of null *Serpinf1^−/−^* heterozygous *Serpinf1^+/−^* and wild type *Serpinf1*^+/+^ mice were *rd8* free (as described before [16]). These mice were maintained in the animal house facility of the National Institutes of Health, and all the experimental procedures were approved by the National Eye Institute Animal Care and Use Committee and performed following the guidelines of the Association for Research in Vision and Ophthalmology statement for the Use of Animals in Ophthalmic and Vision Research, USA. All the experimental animals were maintained on normal chow diet and were kept at 280–300 lux light intensity and 12 h light/12 h dark cycle. Mouse genotyping was provided by an automated service from Transnetyx Inc., and it was confirmed by mouse tail snips genotyping using PCR primers and a protocol previously described [16].

### 4.2. RPE/Choroid Flat Mounts

Eyes from *Serpinf1*^+/+^, *Serpinf1^+/−^* and *Serpinf1^−/−^* mice of 3 months of age were enucleated, and the neural retinas were manually separated from the eyecups containing RPE, choroid and sclera under a dissection microscope using surgical forceps, as previously described [16].

### 4.3. Immunofluorescence and Fluorescent Confocal Microscopy

The RPE/choroid flat mounts were fixed with 4% paraformaldehyde (PFA, Electron Microscopy Sciences, Hatfield, PA, USA, catalog number 15710) in phosphate buffered saline (PBS, GIBCO, Waltham, MA, USA, catalog number 70011-044) for 30 min at room temperature. Then they were washed for 5 min with PBS and the washes were repeated three times. Finally, they were permeabilized with 2% Triton X-100 (SIGMA-ALDRICH, Saint Louis, MO, USA, catalog number T8787-100ML) in PBS for 30 min at room temperature.

For nuclear volume calculations, the permeabilization solution was washed with PBS (as described above) and the RPE/choroid flat mounts were incubated with a PBS solution containing 10 µg/mL Hoechst 33258, pentahydrate (bis-Benzimide) (Thermo Fisher, Waltham, MA, USA, catalog number H3569) and 8.0 U/mL Alexa Fluor™ 488 Phalloidin (Thermo Fisher, Waltham, MA, USA, catalog number A12379) for 30 min in the dark, followed by washes with PBS, as described above. The tissues were mounted on microscope slides for imaging with confocal microscopy to detect fluorescence. The acquired images were used to measure nuclear diameter using ImageJ [44] and to calculate the volume of nuclei using GraphPad (version 8.0.0 for Windows, GraphPad Software, San Diego, California USA, www.graphpad.com). In the GraphPad, radius (r) was calculated by dividing the diameter by 2 (Analyze > Transform > Transform Y values > r = Y/2) and nuclear radius values were used to calculate the nuclei volumes by applying the formula: V = (4/3) πr^3^. For nucleoli quantification, confocal micrographs from RPE/choroid whole mount immunofluorescence were color separated and the nuclei channel (Hoechst) was converted into black and white images using ImageJ. A total of 3 regions within each acquired image were randomly selected, and the punctuated Hoechst staining (nucleoli) was manually counted in 5 nuclei in each region using ImageJ.

For immunodetection of rhodopsin, RPE/choroid flat mounts were fixed and permeabilized as described above and incubated with blocking solution (10% normal goat serum, Thermo Fisher, Waltham, MA, USA, catalog number 50062Z) for an hour at room temperature. An aliquot of anti-rhodopsin antibody (Abcam, Cambridge, MA, USA, catalog number ab230692) was added to the blocking solution to a final concentration of 1 µg/mL and the flat mounts were incubated for an hour at room temperature with gentle rocking. The primary antibody solution was removed, and the RPE/choroid flat mounts were washed with PBS, as described above, followed by incubation of the flat mounts in 2 µg/mL of goat anti-mouse-IgG labeled with Alexa-Fluor-488 dye (Thermo-Fisher, Waltham, MA, USA, catalog number A28175), 10 µg/mL of Hoechst DNA dye and 8.0 U/mL of Alexa Fluor™ 488 phalloidin in blocking solution and for 30 min with gentle rocking at room temperature, and in the dark. For immunodetection of PEDF-R, an anti-PEDFR antibody (Cell Signaling Technology, Danvers, MA, USA, catalog number 2439S) was used following the same protocol described above, diluted to a final concentration of 5 µg/mL, and using a goat anti-rabbit-IgG antibody labeled with Alexa-Fluor-568 dye (Thermo-Fisher, Waltham, MA, USA, catalog number A11036) diluted to a final concentration of 10 µg/mL.

For lipid detection, the RPE/choroid flat mounts were incubated with a solution of 1 µg/mL of BODIPY 493/503 (4,4-Difluoro-1,3,5,7,8-Pentamethyl-4-Bora-3a,4a-Diaza-s-Indacene, Thermo Fischer, Waltham, MA, USA, catalog number D3922) in blocking solution and at room temperature for 30 min and keeping them in the dark. The RPE/choroid flat mounts were washed with PBS as described above and mounted on microscope slides with a one well Secure-Seal™ Spacer of 13 mm diameter and 0.12 mm deep (Thermo Fisher, Waltham, MA, USA, catalog number S24735), 20 µL of mounting medium (ProLong™ Gold Antifade Mountant, Thermo Fisher, Waltham, MA, USA, catalog number P36930) and covered with a glass coverslip (Fisher-Scientific, catalog number 15-183-90).

### 4.4. Fluorescent Imaging Quantification

Imaging was performed using a Zeiss LSM880 confocal microscope with an Airy Scan detector in super resolution mode (SRM), with a 40× or 63× objectives and a 2–3× optical zoom. A region of interest (ROI) was identified about the equatorial region of the RPE in the flat mounts. A total of 3 ROIs per flat mount were identified from 3 animals per genotype. Image stacks of 1 µm thickness were acquired to generate projected images using the Zeiss ZEN Digital Imaging for Light Microscopy software (RRID:SCR_013672).

The fluorescence in each projected fluorescent image was quantified using ImageJ [51]. For rhodopsin quantification, discrete rhodopsin-labeled particles were quantified as follows: the images were color-split, the particle edges were smoothed using a Gaussian blur process (sigma radius: 2), the background was subtracted (using rolling ball radius of 10 pixels), the image threshold automatically adjusted and applied, and rhodopsin particles were automatically counted using Analyze Particles. Rhodopsin particles were normalized to the total number of RPE cells per region of interest (ROI). The total number of cells per ROI was determined by manually counting the RPE cells detected by F-actin (phalloidin) staining. For lipid quantification (BODIPY fluorescence), the corresponding projected fluorescent images were color-split and the total fluorescence per ROI quantified using ImageJ. The fluorescence intensity of BODIPY was normalized to the total number of RPE cells per ROI using F-actin (phalloidin) staining, as described above. For PEDF-R quantification, the corresponding projected fluorescent images were color-split and the total fluorescence per ROI quantified using ImageJ and normalized to total number of RPE cells per ROI.

### 4.5. Transcript Levels Determination by Quantitative PCR (qPCR)

Dissected eyecups from wild-type *Serpinf1*^+/+^, heterozygous *Serpinf1^+/−^* and null *Serpinf1^−/−^* mice were collected, and RNA was isolated using reagents from the RNeasy Mini Kit (Qiagen, Germantown, MD, USA, catalog number 74104) according to manufacturer specifications. RNA from three replicate animals per genotype was isolated independently. The concentration of RNA in the samples was determined using a Nanodrop spectrophotometer (Thermo Fisher, Waltham, MA, USA, catalog number ND-ONE-W). Aliquots of RNA (300 ng) were used for reverse transcription (RT) reactions using SuperScript™ III First-Strand Synthesis System (Thermo Fisher Scientific, Waltham, MA, USA, catalog number 18080051) following the manufacturer’s specifications. The final volume of each RT reaction was adjusted to 300 µL to generate cDNA stock solutions at 1 ng/µL and stored at −80 °C until further utilization.

DNA amplification using quantitative polymerase chain reaction (qPCR) was performed using 0.15 ng/µL cDNA and 0.5 µM of each the forward and reverse PCR primers (see Table 1) with QuantiTect SYBR Green PCR Kit (200) (QIAGEN, Germantown, MD, USA, catalog number 204143) per reaction and following the manufacturer’s specifications. The qPCR reactions were performed in triplicate using a MicroAmp™ Fast Optical 96-Well Reaction Plate with Barcode (Thermo Fisher Scientific, Waltham, MA, USA, catalog number 4346906). The qPCR cycling conditions were: 95 °C (15 min), followed by 35 cycles of 94 °C (15 s), 65 °C (30 s) and 72 °C (30 s) and a melting curve analysis, using a Quant Studio 7 Flex thermocycler (Applied Biosystems, Waltham, MA, USA). The cDNA amplification signals were collected and converted into threshold cycle numbers (CT). The CT numbers were used to calculate the relative gene expression of the selected genes compared to a housekeeping control gene (*Rplp0* encoding for a ubiquitous ribosomal protein) using the delta-delta-CT approach [52].

### 4.6. Western Blot

RPE/choroid tissues dissected from mice were individually transferred into 1.7 mL Eppendorf tubes containing 200 µL of RIPA lysis buffer (VWR Life Sciences, Radnor, PA, USA, catalog number N653-100ML). The lysates were sonicated using a Fisherbrand™ Model 50 Sonic Dismembrator (Thermo Fisher Scientific, Waltham, MA, USA, catalog number, FB50110) setting the amplitude to 30 for 30 s on ice, and then subjected to centrifugation at 20,000× *g* for 30 s at 4 °C and immediately frozen until further use. The total protein in the lysates was quantified using a BCA kit (Pierce™ BCA Protein Assay Kit, Thermo Fisher Scientific, Waltham, MA, USA, catalog number: 23227), following the specifications of the manufacturer and using a SpectraMax iD5 plate reader (Molecular Devices, San Jose, CA, USA, catalog number ID5-STD). Aliquots from the protein lysates containing a total of 5 µg of protein were mixed with NuPAGE™ LDS Sample Buffer (4×) (Thermo Fisher Scientific, Waltham, MA, USA, catalog number: NP0007) containing 5mM dithiothreitol, heated at 95 °C for 5 min, immediately cooled in ice. The loading samples were prepared following specifications for SDS-PAGE electrophoresis using NuPAGE 4–12% polyacrylamide gradient in Bis-Tris gels (Thermo Fisher Scientific, Waltham, MA, USA, catalog number NP0321) and with MOPS SDS Running Buffer (Thermo Fisher Scientific, Waltham, MA, USA, catalog number NP0001). A molecular weight marker (SeeBlue Plus2 Pre-stained Protein Standard, Thermo Fisher Scientific, Waltham, MA, USA, catalog number LC5925) was used for monitoring protein migration during electrophoresis and for protein transfer onto membranes, as well as a reference for protein size. The final volume of the loading samples was 40 µL. After loading the samples into the gels, electrophoresis was at 50 V for 1 h, followed at 100 V for 2 h. After electrophoresis, the proteins in the gels were transferred onto nitrocellulose membranes using iBlot™ 2 Transfer Stacks (Thermo Fisher Scientific, Waltham, MA, USA, catalog number IB23001) and the iBlot 2 Dry Blotting System (Thermo Fisher Scientific, Waltham, USA, catalog number IB21001), following the manufacturer’s specifications. Detection of proteins was performed using LI-COR reagents as it follows. The nitrocellulose membranes were washed with PBS (3 times) and incubated with 5 mL of REVERT™ 700 Total Protein Stain solution (Li-COR, Lincoln, NE, USA, catalog number 926-11021), in the dark, for 5 min at room temperature. The membranes were imaged using an Odyssey^®^ DLx imaging detector (LI-COR) following company’s specification (https://www.licor.com/documents/u4nl1wri6odkz2vwhvwlu6uaziw94btu (accessed on 20 February 2022)). The nitrocellulose membranes were rinsed with PBS, washed with 5 mL of REVERT™ 700 WASH solution (LI-COR, Lincoln, NE, USA, catalog number 926-11022) and incubated with 5 mL of blocking solution (Intercept Protein-Free Blocking Buffer, LI-COR, Lincoln, NE, USA, catalog number 927-90001, diluted 1:1 in PBS) for 2 h at room temperature. An anti-PEDF antibody (Rabbit Anti-PEDF polyclonal antibody, XpressBio, Frederick, MD, USA, catalog number AB-mPEDF1, 250 ng/mL, diluted in blocking solution) was added to the nitrocellulose membranes and incubated at 4 °C for 16 h in a rocking platform. The membranes were washed 3 times with PBS-T (PBSs containing 0.1% Tween-20 (Millipore-Sigma, Saint Louis, MO, USA, catalog number P1379) for 5 min each time and incubated with a secondary antibody (IRDye^®^ 800CW Donkey anti-Rabbit IgG Secondary Antibody, LI-COR, Lincoln, NE, catalog number 926–32213, at 1 µg/mL) in the blocking solution for 1 h in the dark, at room temperature. The membranes were finally washed in PBS-T for 5 min, in the dark, at room temperature and repeated 3 times. Images were acquired using the Odyssey^®^ DLx imaging detector.

### 4.7. β-Galactosidase Activity Assay

This assay was adapted from the literature [30] as follows. Eyecups from mice were dissected and each RPE/choroid was transferred to an individual well of a 24-well cell culture plate (Fisher Scientific, catalog number FB012929) containing 0.5 mL of culture medium consisting of Dulbecco’s Modified Eagle’s Medium (DMEM, Thermo Fisher Scientific, Waltham, MA, USA, catalog number 12100-046), 2% fetal bovine serum (FBS, Thermo Fisher Scientific, Waltham, MA, USA, catalog number 10082147) and 10.0 U/mL penicillin/streptomycin (Thermo Fisher Scientific, Waltham, MA, USA, catalog number 15140122). The fluorescently labeled substrate for β-Galactosidase, C_12_FDG (5-dodecanoylaminofluorescein di-β-D-galactopyranoside, Thermo Fisher Scientific, Waltham, MA, USA, catalog number D2893) was added to each well to a final concentration of 0.66 µM. The RPE/choroid flat mounts were incubated in the dark at 37 °C for 1.5 h. Then they were fixed with PFA, permeabilized and stained with Hoechst and phalloidin as described above under the session Fluorescent Confocal Microscopy. The RPE/choroid flat mounts were mounted on glass slides and the images were acquired using a Zeiss LSM880 confocal microscope as described above under Immunofluorescence Imaging and Quantification. The quantification of the fluorescent product 5-DAF (5-Dodecanoylaminofluorescein) resulting from the β-Galactosidase activity was performed using projected fluorescent images and ImageJ. The density intensity of 5-DAF was normalized to the total number of RPE cells per ROI using F-actin (phalloidin) staining as described above.

### 4.8. Statistical Analysis

The statistical analysis was performed using PRISM-GraphPad 9.1.2 software. Student’s T-Test or non-parametric ANOVA were employed to compare 2 data points or more than 2 data points, respectively.

## Figures and Tables

**Figure 1 ijms-23-07745-f001:**
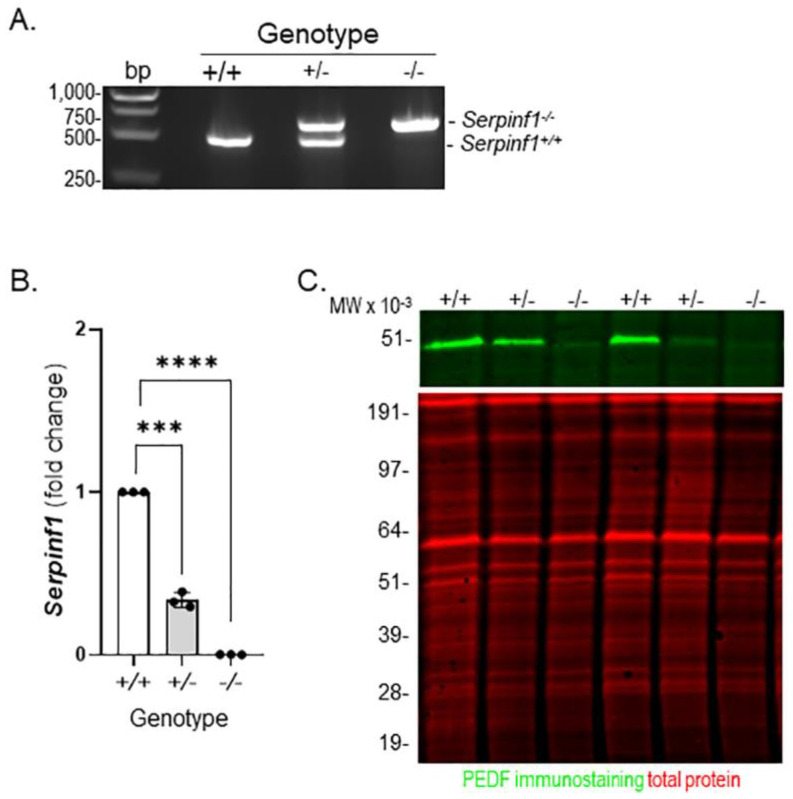
*Serpinf1* deletion in mouse RPE. (**A**) A photo of an agarose gel of PCR reactions using DNA from mouse tail snips for genotyping is shown. Lanes contained samples as indicated at the top for: DNA markers (bp); PCR reactions with DNA from *Serpinf1^+/+^* (+/+), *Serpinf1^+/−^* (+/−) and *Serpinf1^−/−^* (−/−). Migration pattern of the DNA markers, and bands of the PCR products are indicated to the left and right sides of the photo, respectively. (**B**) Transcript level determination of the *Serpinf1* gene in RPE/choroid flat mounts from *Serpinf1^+/+^*, *Serpinf1^+/−^* and *Serpinf1^−/−^* mice detected by quantitative PCR (qPCR) was performed in triplicate from three mice per genotype. The plot shows fold change of transcripts relative to the levels of *Serpinf1^+/+^* in the *y*-axis and the genotype in the *x*-axis with individual data points per animal. Statistical analysis: *** *p* < 0.001, **** *p* < 0.0001. (**C**) Photos of a Western blot of RPE/choroid protein lysates prepared from two animals for each *Serpinf1^+/+^*, *Serpinf1^+/−^* and *Serpinf1^−/−^* mice are shown. Protein samples loaded to each lane are indicated at the top. The upper image corresponds to the immunodetection of PEDF protein (green) using a specific PEDF antibody in the western and the lower image corresponds to total protein staining (red). Migration pattern of the protein molecular weight standards is indicated to the left.

**Figure 2 ijms-23-07745-f002:**
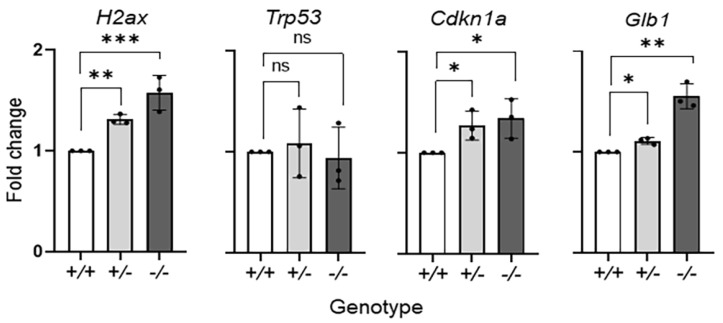
*Serpinf1* deletion increased senescence-associated gene expression in the RPE. Transcript level determination of senescence-associated genes *H2ax*, *Trp53*, *Cdkn1a* and *Glb1* in RPE/choroid flat mounts from *Serpinf1^+/+^*, *Serpinf1^+/−^* and *Serpinf1^−/−^* mice detected by quantitative PCR (qPCR) was performed in triplicate from three mice per genotype. The plot shows fold change of transcripts relative to the levels of *Serpinf1^+/+^* in the *y*-axis and the genotype in the *x*-axis with individual data points per animal. Statistical analysis: * *p* < 0.01; ** *p* < 0.001; *** *p* < 0.001).

**Figure 3 ijms-23-07745-f003:**
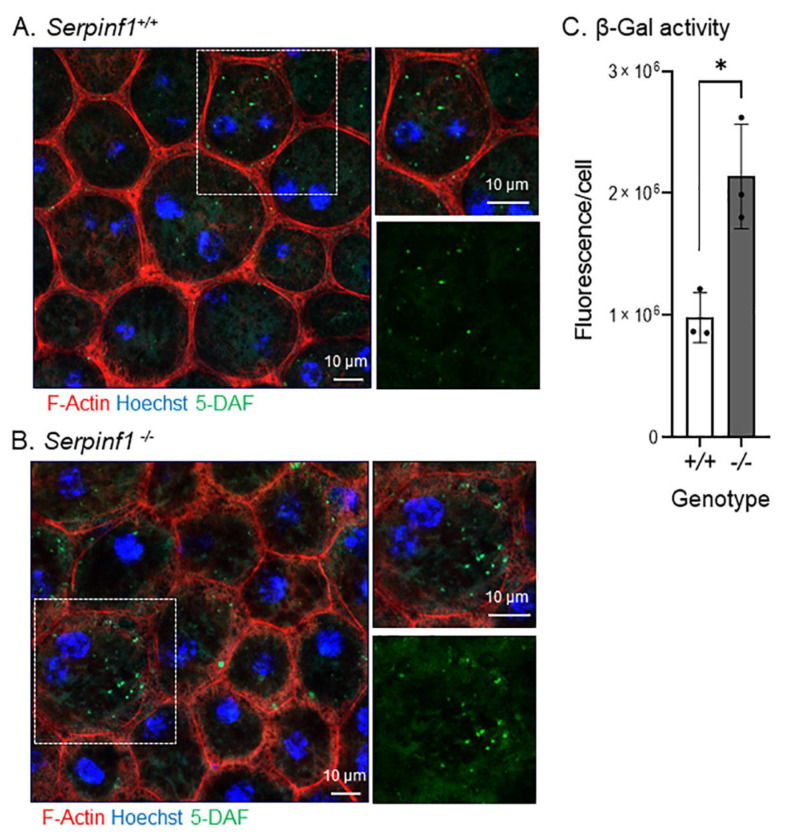
*Serpinf1* deletion increased senescence-associated β-galactosidase (SA-β-gal) activity in the RPE. (**A,B**) Representative microphotographs of the in-situ SA-β-gal enzymatic activity detected in RPE/choroid flat mounts from *Serpinf1^+/+^* (**A**) and from *Serpinf1^−/−^* (**B**) mice. The substrate of the reaction was a SA-β-gal substrate (C_12_FDG). The resulting products of SA-β-gal enzymatic (5-DAF), phalloidin and Hoechst staining were detected by fluorescent confocal microscopy. Magnification of an area marked by a dotted square is shown on the right side for the overlay (top), and 5-DAF only (bottom). (**C**) The plot shows 5-DAF fluorescence per cell from the average of 3 ROIs per flat mount (each data point) and from 3 mice per genotype. * *p* < 0.01.

**Figure 4 ijms-23-07745-f004:**
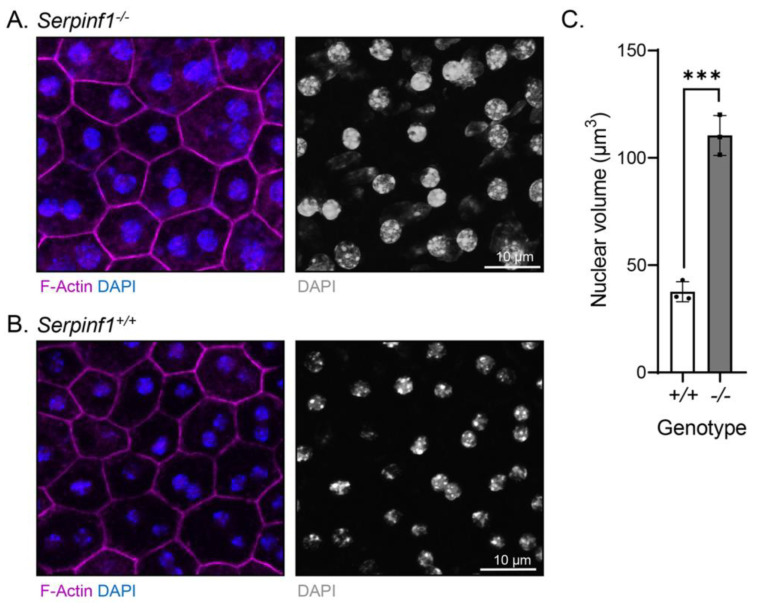
*Serpinf1* deletion caused nuclear size differences in the RPE. (**A**,**B**) Representative images acquired from ROIs of RPE/choroid flat mounts of *Serpinf1^−/−^* (**A**) and *Serpinf1^+/+^* (**B**) mice stained with DAPI (blue) and phalloidin (magenta) using fluorescence and confocal microscopy. Photomicrographs of an overlay of DAPI and phalloidin are shown on the left side, and of only DAPI in black and white on the right side, as indicated below each image. (**C**) The diameter of the nuclei determined from images as in panel A and B with ImageJ was used to calculate the nuclear volume and plotted using GraphPad. The plot shows nuclear volumes from the average of 3 regions of interest (ROIs) per flat mount (each data point) and from 3 mice per genotype. The average volume of *Serpinf1^+/+^* RPE was 37.6 µm^3^, and of *Serpinf1^−/−^* RPE was 110.4 µm^3^ (*** *p* < 0.001).

**Figure 5 ijms-23-07745-f005:**
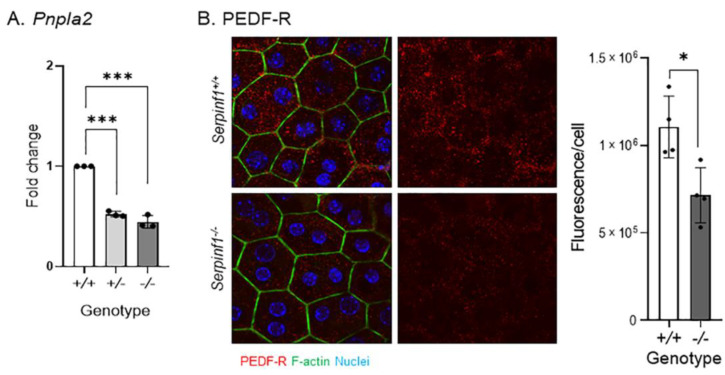
(**A**) Transcript level determination of the *Pnpla2* gene in RPE from *Serpinf1^+/+^*, *Serpinf1^+/−^* and *Serpinf1^−/−^* mice detected by qPCR was performed in triplicate from three mice per genotype. The plot shows fold change of transcripts relative to the levels of *Serpinf1^+/+^* in the *y*-axis and the genotype in the *x*-axis with individual data points per animal. Statistical analysis: *** *p* < 0.001. (**B**) Protein level determination of the PEDF-R in the RPE from *Serpinf1^+/+^* and *Serpinf1^−/−^* mice detected by immunofluorescence was performed using 4 regions of interest per genotype. The representative micrographs on the left show PEDF-R detection in the *Serpinf1^+/+^* and *Serpinf1^−/−^* RPE (upper and lower images, respectively). The graph on the right corresponds to the quantification of PEDF-R immunofluorescence normalized to the total number of RPE cells per ROI. Statistical analysis: * *p* < 0.01.

**Figure 6 ijms-23-07745-f006:**
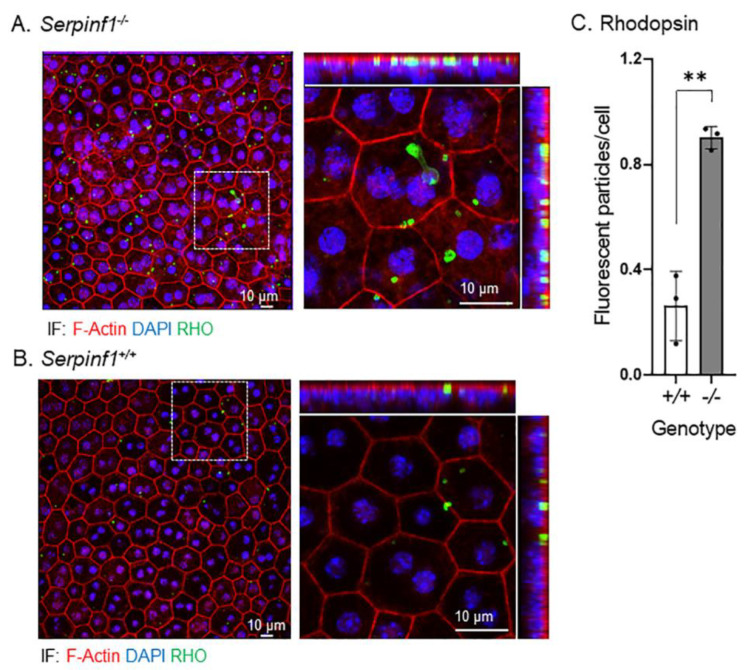
Rhodopsin accumulated in the *Serpinf1^−/−^* RPE. (**A**,**B**) Representative images of ROIs of RPE/choroid flat mounts from *Serpinf1^−/−^* (**A**) and *Serpinf1^+/+^* (**B**) mice immunostained with anti-rhodopsin (green) and stained with DAPI (blue) and phalloidin (magenta) using immunofluorescence are shown. The confocal projected images were magnified (see the dotted square) and shown on the right with X-Y, X-Z and Y-Z projections. (**C**) The plot shows fluorescence of anti-rhodopsin per cell from the average of 3 ROIs per flat mount (each data point) and from 3 mice per genotype. ** *p*< 0.001.

**Figure 7 ijms-23-07745-f007:**
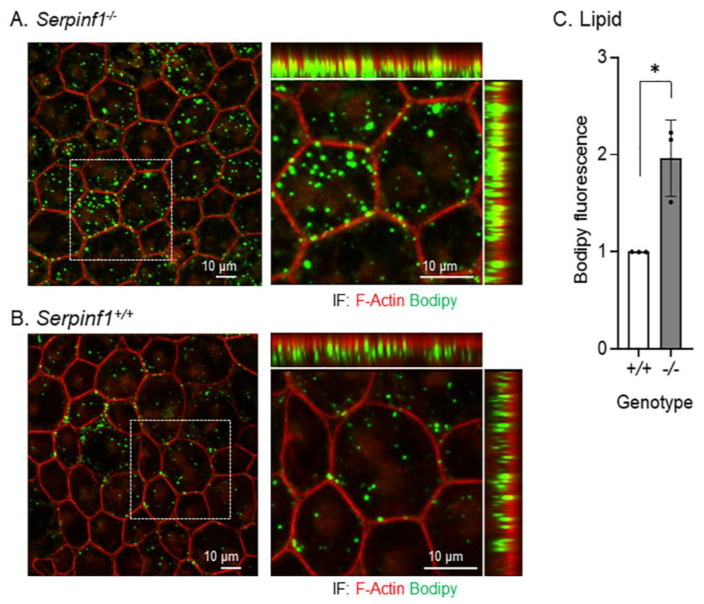
Lipids accumulated in the *Serpinf1*-depleted RPE. (**A**,**B**) Representative images of ROIs of RPE/choroid flat mounts from *Serpinf1^−/−^* (**A**) and Serpinf1^+/+^ (**B**) mice stained with BODIPY (green), DAPI (blue) and phalloidin (magenta) using immunofluorescence are shown. The confocal projected images were magnified (see the dotted square) and shown on the right with X-Y, X-Z and Y-Z projections. (**C**) The plot shows BODIPY fluorescence per cell from the average of 3 ROIs per flat mount (each data point) and from 3 mice per genotype. * *p* < 0.01.

**Table 1 ijms-23-07745-t001:** Sequences for qPCR primers.

Gene	Protein	Forward Primer	Reverse Primer
*Rplp0*	RPLP0	5′-CTTCATTGTGGGAGCAGACA-3′	5′-GTGAGGTCCTCCTTGGTGAA-3′
*Serpinf1*	PEDF	5′-ACCGTGACCCAGAACTTGAC-3′	5′-CACGGGTTTGCCAGTAATCT-3′
*Pnpla2*	PEDF-R	5′-ACAGTGTCCCCATTCTCAGG-3′	5′-TTGGTTCAGTAGGCCATTCC-3′
*H2ax*	H2AX	5′-GCCTCATACCAGTTGACCCTG-3′	5′-TAGAACTCTTGTCCACAGGCC-3′
*Trp53*	P53	5′-GACCCTGGCACCTACAATGAA-3′	5′-GGGGTGGATAAATGCAGACA-3′
*Cdkn1a*	P21	5′-ATCTGCTGCTCTTTTCCCCC-3′	5′-CCCTAGACCCACAATGCAGG-3′
*Glb1*	GLB1	5′-GCACGGCATCTATAATGTCACC-3′	5′-AGCCGGTCCTCCCAGTAG-3′

## Data Availability

Not applicable.

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
