# Peer review of "PEDF Deletion Induces Senescence and Defects in Phagocytosis in the RPE"

_ijms, 2022, doi:10.3390/ijms23147745_

Round 1

Reviewer 1 Report

This manuscript by Rebustini et al. investigates the role of PEDF loss in senescence and phagocytosis, supporting its dual role in the homeostasis of RPE cells. Authors demonstrate that Serpinf1 deletion results in increased senescence-associated gene expression, phagocytic dysfunction leading to accumulation of rhodopsin and lipid deposits, mediated through reduced PEDF-R expression. The manuscript is very well written. I have some general comments. 

1. One concern is that there might be a range of mechanisms that would result in the induction of senescence. Although this is quite interesting, it is worth considering classical negative markers of cellular senescence like Cyclin A2, Cyclin B1 and Cyclin D1, to confirm what is necessary for senescence.

2. What is the expression pattern of senescence markers in mice <3 months old? Does absence of PEDF affect the morphology of RPE, nucleoli number and nuclei size in these mice?

3. Line 210-211; should be read as- “The plot of the number of rhodopsin-stained particles per cell showed that the Serpinf1 -/-  RPE accumulated RHO 3.5-fold than in the Serpinf1 +/+  RPE (Fig. 6C) and not “The plot of the number of rhodopsin-stained particles per cell showed that the Serpinf1 -/-  RPE accumulated RHO 3.5-fold than in the Serpinf1 -/-  RPE (Fig. 6C)”

4. The PEDF absence resulted in disorganized distribution pattern of F-actin and other phenotypic changes in RPE.  The interpretation sounds logical and it could be incorporated within the discussion

Author Response

In addition to point-by-point responses (see attached file and here below), we have corrected italics for gene names in the manuscript.  

Reviewer 1

Comments and Suggestions for Authors

This manuscript by Rebustini et al. investigates the role of PEDF loss in senescence and phagocytosis, supporting its dual role in the homeostasis of RPE cells. Authors demonstrate that Serpinf1 deletion results in increased senescence-associated gene expression, phagocytic dysfunction leading to accumulation of rhodopsin and lipid deposits, mediated through reduced PEDF-R expression. The manuscript is very well written. I have some general comments.

We thank the reviewer for the kind words about our manuscript.

  1. One concern is that there might be a range of mechanisms that would result in the induction of senescence. Although this is quite interesting, it is worth considering classical negative markers of cellular senescence like Cyclin A2, Cyclin B1 and Cyclin D1, to confirm what is necessary for senescence.

Indeed, there might be a range of mechanisms that would result in the induction of senescence, and this is of great interest. The recommendation to cyclin A2, Cyclin B1 and Cyclin D1 as negative markers to confirm if they are involved in senescence is of interest. However, given the deadline of July 5th, we do not have time to complete the experiment, i.e., a western blot with retinal extracts and antibody to these proteins. This will be included in a follow up study on the topic. In addition, our choice of positive markers of senescence tend to be more conclusive than negative markers that decrease senescence.

  1. What is the expression pattern of senescence markers in mice <3 months old? Does absence of PEDF affect the morphology of RPE, nucleoli number and nuclei size in these mice?

As mentioned in our response above, the deadline of July 5th precludes our performing the experiment suggested by the reviewer and will be considered for a future study.

  1. Line 210-211; should be read as- “The plot of the number of rhodopsin-stained particles per cell showed that the Serpinf1 -/- RPE accumulated RHO 3.5-fold than in the Serpinf1 +/+ RPE (Fig. 6C) and not “The plot of the number of rhodopsin-stained particles per cell showed that the Serpinf1 -/-  RPE accumulated RHO 3.5-fold than in the Serpinf1 -/-  RPE (Fig. 6C)”

The sentence has been corrected and is now in lines 234-235.  Thank you for noticing this typo error.

  1. The PEDF absence resulted in disorganized distribution pattern of F-actin and other phenotypic changes in RPE. The interpretation sounds logical and it could be incorporated within the discussion.

Thank you again, and the following has been inserted in the first paragraph of the Discussion and is in lines 266-267:“; and 4) disorganized distribution pattern of F-actin and other phenotypic changes”

Reviewer 2 Report

The manuscrip describes that PEDF loss as a cause of senescence-like changes in the RPE to point out a retinoprotective and a regulatory protein of aging-like changes associated with defective degradation of photoreceptor outer segment in the RPE.

Please take into account the following suggestions:

1-      Add more references in the introduction about lines of evidence that suggest that lipid metabolism and lipid reprogramming play an important role in aging, longevity, and age-related diseases and explain this in more detail.

2-      Explanation for Fig 1, 3 is too long, please add part  the info in the main text

3-      Discuss why authors think that ……….This is the first time that the Serpinf1-/- mouse has been used to study senescence and phagocytosis of the RPE.

Author Response

In addition to point-by-point responses (Please see the attachment and here below), we have corrected italics for gene names in the manuscript.  

Reviewer 2

Comments and Suggestions for Authors

The manuscrip describes that PEDF loss as a cause of senescence-like changes in the RPE to point out a retinoprotective and a regulatory protein of aging-like changes associated with defective degradation of photoreceptor outer segment in the RPE.

Please take into account the following suggestions:

1-      Add more references in the introduction about lines of evidence that suggest that lipid metabolism and lipid reprogramming play an important role in aging, longevity, and age-related diseases and explain this in more detail.

We thank the reviewer for the suggestion of adding more information. Most of the lines of evidence are cited in the references of the cited papers. Nevertheless, we have added lines of evidence with explanation about some of the roles of lipid metabolism and reprogramming in aging, longevity and age-related diseases in the Introduction:

Plasma lipidomic profiles of 11 different mammalian species with longevities varying from 3.5 to 120 years can accurately predict the lifespan of animals and, in particular, plasma long-chain free fatty acids and lipid peroxidation-derived products are inversely correlated with longevity [22]. Similarly, the genomes of 25 different species reveal that genes involved in lipid composition undergo increased selective pressure in longer-lived animals [23]. Evidence from animals with extreme longevity also links lipid metabolism to aging. The ocean quahog clam Arctica islandica, an exceptionally long-lived animal that can survive for more than 500 years, exhibits a unique resistance to lipid peroxidation in mitochondrial membranes [24]. The bowhead whale, another complex animal with extreme longevity that can live longer than 200 years, has lens membranes that are especially enriched with phospholipids. This unique enrichment is thought to partially underlie its uncanny resistance to the age-related lens disease of cataracts [25]. Naked mole rats, which enjoy remarkably long lifespans for rodents, have a unique membrane phospholipid composition that has been theorized to contribute to their exceptional longevity [26]. The importance of lipids in lifespan is further confirmed by the ability of lipid-related interventions to enhance longevity in model organisms [27]. It has been reported that senescent cells accumulate lipid droplets and import lipid tracers and that treating proliferating cells with specific lipids induces senescence, placing deregulation of lipid metabolism as a factor to regulate cellular senescence [20].

There are also pathologies and conditions associated with altered nuclear shape in which lipid modification of lamin A causes for premature aging, and also normal aging is associated with abnormal nuclear shape, and linked to lipid modifications of nuclear lamin, and progerin [28-29]. Interestingly, nuclear shape can be affected by lipid synthesis, e.g, inactivation of lipid phosphatase (lipin) causes expansion and alteration of nucleus phospholipid synthesis and expanding nucleus [28-29].” 

2-      Explanation for Fig 1, 3 is too long, please add part  the info in the main text

Part of the information is in the main text.

3-      Discuss why authors think that ……….This is the first time that the Serpinf1-/- mouse has been used to study senescence and phagocytosis of the RPE.

We assert that this is the first report on the use of the Serpinf1-/- mouse to study the impact of PEDF depletion in senescence and phagocytosis of the RPE from our labs.  PUBMED does not show a report(s) that precedes this study by other research group(s).

Reviewer 3 Report

I read the paper entitled  “PEDF deletion induces senescence and defects in phagocytosis in the RPE ” very carefully and concluded that the paper is acceptable with minor revision in the present form for publication in your journal.

Figure 2 is not included in the text and must be added.

The topic of the article is interesting. The role of RPE in phagocytosis in neural retina is known as one of the most active phagocytosis in the body. This study contributes to importance of PEDF signaling of age-related processes and and its role to RPE homeostasis.

At the end I would like to thank you considering me as a reviewer for the report.

Author Response

In addition to point-by-point responses (Please see the attachment and here below), we have corrected italics for gene names in the manuscript.  

Reviewer 3

Comments and Suggestions for Authors

I read the paper entitled  “PEDF deletion induces senescence and defects in phagocytosis in the RPE ” very carefully and concluded that the paper is acceptable with minor revision in the present form for publication in your journal.

Figure 2 is not included in the text and must be added.

With all due respect, but the reviewer missed that ‘Figure 2’ is mentioned in the Results section in line 132 and as caption in lines 140-145.

The topic of the article is interesting. The role of RPE in phagocytosis in neural retina is known as one of the most active phagocytosis in the body. This study contributes to importance of PEDF signaling of age-related processes and and its role to RPE homeostasis.

We thank this reviewer for the kind words about the study described in our manuscript.

At the end I would like to thank you considering me as a reviewer for the report.
